# Achieving Neighborhood-Level Collaborative Governance through Participatory Regeneration: Cases of Three Residential Heritage Neighborhoods in Shanghai

**Han Wang [1,†] and Yueli Xu [2,*,†]**

[1] College of Fashion and Design, Donghua University, Shanghai 200051, China; wanghan@dhu.edu.cn
[2] Department of Urban Planning and Design, Design School, Xi'an Jiaotong–Liverpool University, Suzhou 215123, China
[*] Correspondence: yuelixu.pt@xjtlu.edu.cn
[†] These authors contributed equally to this work.

**Abstract:** Residential heritage neighborhoods in China are experiencing a transformation from large-scale demolishment, which is associated with relocating residents, to small-scale neighborhood regeneration, and distinctive models of participatory regeneration are emerging. Participatory regeneration is increasingly considered to be an effective way to achieve multiple goals in urban development; however, little research has investigated the impacts of participatory regeneration on achieving neighborhood-based collaborative governance. This study aims to explore which mechanisms associated with participatory regeneration facilitate or constrain neighborhood-based collaborative governance, using case studies in Shanghai. Based on the investigation of three nuanced pathways in participatory regeneration, this study compares four dimensions associated with participatory regeneration (i.e., participatory decision-making, inclusion, collective problem-solving, and power-sharing) and explores key mechanisms that are applicable for achieving collaborative governance in different scenarios, aiming to enhance social development and social sustainability in future. The findings indicate that although grassroots government played vital roles in participatory heritage neighborhood regeneration through financial support and supervision strategies, facilitating mechanisms could still be observed for achieving collaborative governance. Furthermore, this study provides suggestions for planners in building collaborative governance in other developing areas which are experiencing rapid urbanization with strong state interventions.

**Keywords:** collaborative governance; participatory neighborhood regeneration; participatory decision-making; inclusion; collective problem-solving; power-sharing





## 1. Introduction

Sustainability has inspired fervent discussions since 1987 and has gradually developed into an inclusive and integrated concept, covering environmental, economic, and social dimensions [1].Since inappropriate urban renewal tends to adversely affect living environments, via methods such as demolishing buildings and relocating residents, sustainable urban development is essential for protecting well-being and vitality [2]. When issues of large-scale renewal under rapid urbanization were increasingly acknowledged, contemporary urban regeneration started to transform from technical space planning into comprehensive urban governance, and its aim started to transform from physical upgrading or economic gentrification to neighborhood-level sustainable development [3]. Participation and collaborative governance are emphasized as important for making important strategic choices and improving urban sustainability, owing to their provisions of innovative local knowledge and negotiation mechanisms in decision-making and problem-solving [4,5].

China has witnessed significant changes in urban landscapes and internal structures since the market-oriented reform in 1978. Owing to the state enterprises (*danwei*) shedding

social welfare responsibilities to the market and neighborhoods, accompanied by the elimination of housing and the loosening of the *hukou* system (a household registration system in mainland China), immigrants started to flow to cities, despite urban housing dramatically lacking at that time [6]. Since then, two main approaches have been adopted to accommodate the rapidly growing numbers of migrants and to improve urban infrastructure, namely urban renewal and urban expansion. As increasing reflection on the weaknesses of urban expansion was undertaken, the central government started to issue policies to protect arable land through strict control, making urban renewal dominant in megacities. China experienced more than two decades of large-scale urban renewal, involving the demolishment of traditional neighborhoods, the relocation of original residents, and the creation of high-end commodity housing and commercial complexes [7]. Criticisms arose, especially regarding the neglect of affected residents' interests, the break-down of existing social networks, and the lack of meaningful participation to protect the interests of low-income people [7]. Under such circumstances, new principles of urban planning were created by the central government in 2001, promoting the abandonment of large-scale demolishment, and New-Type Urbanization was proposed in 2014, emphasizing people-oriented urban development. This was not only because of the increasing reflections on the social issues generated by large-scale urban renewal but also the development of civil society and the citizens' increasing awareness of participation in urban development. Contemporary China is experiencing a transformation from large-scale resident relocation-based urban renewal to small-scale participatory regeneration, as well as the promotion of innovative social governance and sustainability.

The importance of participation in neighborhood regeneration has been a hot topic in urban studies. Participatory regeneration tends to achieve effective and efficient projects, owing to the acquirement of local resources (i.e., skills and knowledge) and the smooth implementation of the projects [8]. Additionally, participatory regeneration provides the legitimacy of state-led projects to achieve "good governance" and empowers residents' rights to identify collective issues [9]. Practically, as a complex process (with potential issues, i.e., benefit gambling and property rights conflicts) and the involvement of various stakeholders, most neighborhood regeneration projects require strong state interventions, leaving little space for residents [10]. This situation is much more serious in heritage neighborhood regeneration projects, which embrace a wide range of vital social, economic, and cultural values.

Although the values of heritage neighborhoods have drawn local and global attention, these residential-use architectures and landscapes are becoming increasingly vulnerable to large-scale urban renewal, leading to most of them having been demolished for economic benefits or inadequately managed in the context of rapid urbanization, owing to a lack of funding and responsible actors. Owing to the tensions between heritage conservation and neighborhood redevelopment, heritage neighborhoods are regarded as typical cases used to discuss the issues and conflicts associated with urban renewal. Participatory regeneration in heritage neighborhoods becomes vital in cases not only of urban renewal but also of social governance in China, which is particularly imperative in Shanghai. Most heritage neighborhoods used to be either regenerated into other functions for commercial use or demolished for high-end residential areas in Shanghai. In the 2010s, as the promotion of participation in urban development and the deepening development of civil society in urban China increased, more and more heritage neighborhood participatory approaches started to be recorded and distributed on civic organization reports, videos, and websites, aiming to "save" heritage neighborhoods [11]. Apart from regarding heritage neighborhoods as "the commons" and conducting collective actions through voluntarism [12], participatory regeneration emphasizes the role of participation approaches as social solutions in urban renewal for the enhancement of neighborly relations [13] and the strengthening of neighborhood's adaptive capacity [14], owing to the strength of participation in framing neighborhood strategies and monitoring local service delivery [15].

In the literature, scholars have examined the relationship between collaborative governance and participatory regeneration. In urban contexts, from the positive side, enabling the role of participatory approaches could contribute to collaboration through creating power-sharing relationships between researchers and communities, enhancing mutual understanding, offering opportunities for underprivileged groups to give them a chance to be involved, and enhancing both interpersonal trust and institutional trust [16]. From the negative side, scholars point out the constraining role of participation in aggravating conflicts through the inclusion of opponents [17]. In the context of urban renewal, some scholars highlight the vital role of collaborative workshops serving as practical solutions for reaching consensus in participatory planning [18,19]; some emphasize that street-level collaborative governance is more effective at dealing with conflicts than macro-level governance, owing to less of a power imbalance and a strong commitment to collaborative processes [20]; some investigate different types of collaborative behavior among stakeholders (e.g., the degree of cooperation among related governments, conflict resolution efficiency, the degree of public participation, and the normality of public participation) associated with participatory regeneration [19]; and some explain the participatory regeneration process through the collaborative governance theory [21]. However, these studies investigate either collaborative behaviors in the participation process or the influencing role of existing collaborative governance in shaping participatory regeneration, leaving investigations into the influencing role of participatory regeneration in shaping future collaborative governance understudied.

To fulfill this gap, this paper hypothesizes that participatory regeneration affects neighborhood-based collaborative governance, and we have developed two major objectives: firstly, to investigate features relating to the conditions for achieving collaborative governance in three nuanced pathways of participatory regeneration projects; and, secondly, to explore facilitating and constraining mechanisms associated with participatory regeneration in shaping conditions for achieving collaborative governance in the future. This study develops a framework of conditions for achieving collaborative governance and contributes to the discussion of the impacts of participation regeneration on collaborative governance based on the provision of empirical cases in Shanghai, aiming to deepen the understanding of participatory regeneration in China's urban heritage neighborhoods under strong state control.

## 2. Conceptual Framework: Key Dimensions for Achieving Collaborative Governance

In the face of continuous criticism of conventional government-led large-scale urban renewal, which is a closed decision-making process and lacks participatory strategies, more efforts have been focused on cultivating a horizontal and inclusive governance mode [22]. Compared with the insufficient funding associated with conventional hierarchical governance, the involvement of market actors can enhance efficiency through building contractual relations with local states for regeneration projects [23]. However, state-dominant and market-dominant modes both lack consideration for the residents' inputs, leading to researchers calling for participatory pathways towards consensus-oriented decision-making through the inclusion of multiple-levels of state, market, and civil society stakeholders [24].

In the context of urban renewal, participatory regeneration refers to diverse interactions between residents and other stakeholders leading to physical environment improvements, emphasizing the transformation of the regeneration project from a purely technical process of planning to an interactive social process of cooperation [25]. In view of regenerating contents and participating strategies, participatory regeneration consists of physical improvements in the built environment, participatory activities associated with regeneration, and institutional arrangements shaping participatory activities. Specifically, physical improvements relate to upgrading the built environment of the physical space. Social participation, serving as one out of two dimensions of public participation (the other aspect is political participation, such as voting), is more common in China's neighborhoods,

referring to a social process of empowering residents to identify issues and making decisions on the improvement of neighborhood resources' distribution arrangements and cultivating resident volunteers and social groups [2,26]. Institutional arrangements refer to the combination of formal and informal settings of participation, through shaping stakeholders' behaviors and conditions on changing decisions [27]. These dimensions construct three pathways to examine the key features of participatory regeneration.

Furthermore, many efforts have focused on the discussion of conditions for achieving collaborative governance, such as ways of making collective decisions through the creation of rules to govern participation behaviors [28], involving multi-level stakeholders in collaboration [24], creating innovative cooperative problem-solving approaches instead of traditional government arrangements [25], and the establishment of shared power structures to achieve consensus [27]. These dimensions of conditions for achieving collaborative governance are further discussed in the following sections.

## 2.1. Participative Decision-Making

Participative decision-making emphasizes that the process of making decisions is generated and achieved through residents and stakeholders' participative behaviors and activities, instead of hierarchical or coalitional politics, which requires all participants to have an awareness of their rights and responsibilities in achieving a consensus on the proposed issues [29]. The achievement of participatory decision-making relies on the lack of authoritative power relations and weak administrative structures, the empowerment of participants, and the horizontality of small bureaucratic controls [30,31]. Participatory decision-making emphasizes that participants have access to the information shared, any comments delivered, the negotiations conducted, and the agreements reached, demonstrating that participants tend to be willing to support and accept the decisions by providing information, distributing benefits, supervising behaviors, and building commitment [12,28], which might, to a large extent, make the deployment of rules visible and facilitate participants' engagement in collaborative governance.

## 2.2. Inclusion

Ascertaining which people should be included and how they should be included is crucial in understanding collaborative governance. A wide range of involvement of different residents' voices and multi-level actors might, to a large extent, increase opportunities for mutual interactions and social participation but may either enhance or constrain the collaborative process [24]. This is because inclusion, on one hand, could foster procedural justice and ensure citizen participation; however, on the other hand, it might increase transaction costs through the involvement of uncooperative participants and the creation of instabilities when trying to achieve consensus [32]. In this regard, the inclusion of participants needs to be strategic and selective; nuanced and detailed inclusion strategies are needed in order to develop a deeper understanding of the participation process and a better understanding of the use of inclusiveness strategies in shaping collaborative governance [33].

## 2.3. Collective Problem-Solving

Participants tend to participate and engage in collaboration when they have a shared vision and commitment levels for a project highly reliant on collective efforts that could hardly be achieved solely, making collective problem-solving an important aspect for understanding collaborative governance [34]. In this regard, participants can be mobilized and facilitated to move forward in problem-solving rather than problem-blaming. It is noted that participants' engagement in collective problem-solving relies on the condition that the collective way of solving one another's issues does not damage their own interests [35]; at the same time, the more resources participants have that are based on others' needs, the more likely they are to collaborate with others in dealing with collective problems. This also echoes some scholars' observations that mutuality, in addressing conflicts among different participants, is the foundational technique for achieving collaboration, from the perspective

of organizational behavior [36]. In addition, commitment, reciprocity, the equal distribution of costs and benefits, and mutual trust all contribute to collective problem-solving, because they reduce complexity and transaction costs when reaching agreements [37].

### 2.4. Power-Sharing

Serving as a vital resource, power could be viewed as an effective tool to facilitate interactions and mutual dependence between public, private, and civil society actors in collaborative governance, owing to its advantages in demonstrating not only formal hierarchical administrative relations or informal dispersing arrangements but also temporary sharing of financial relations among multiple levels of stakeholders [38]. Power-sharing relates to the equilibrium of power arrangements and corresponding responsibilities in a setting of the collaboration environment, which might embrace collaborative governance [39]. As they end neither as static nor in one universal way, power-sharing arrangements might enable the reaching of different general consensuses in different circumstances and, at the same time, might generate contests and conflicts at the margins [39]. In this regard, the observation of power-sharing contingencies is important, and the control of ideas, resources, and rules between multi-level stakeholders could contribute to collaborative governance [40]. In this study, participative decision-making, inclusion, collective problem-solving, and power-sharing are identified as important for achieving collaborative governance, and these served as the framework to explore the extent to which participatory regeneration could affect collaborative governance in the context of urban renewal. The conceptual framework is displayed in Figure 1.

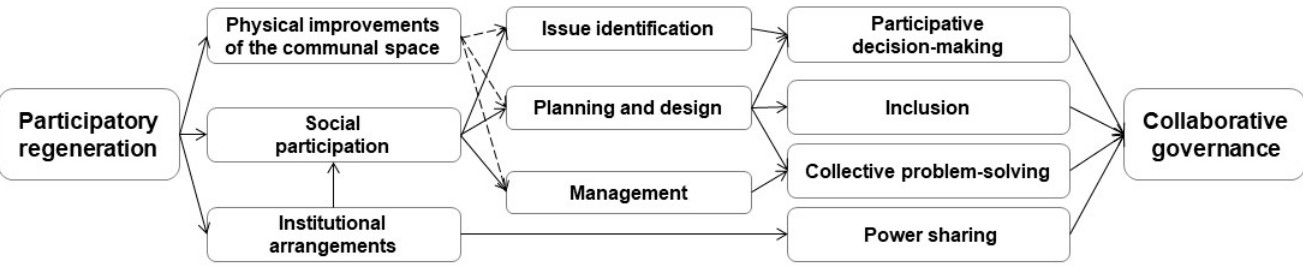

**Figure 1.** Conceptual framework (source: authors).

## 3. Research Design

### 3.1. Case Study

To address the main aim of exploring the influencing role of participatory regeneration in shaping collaborative governance in future, the method employed needs to capture the underlying principles of an occurrence within a real-life context and provide a more in-depth explanation. Thus, this study adopts a case study as the main research method, due to its advantages in the provision of detailed explanation besides only descriptive information, which might generate a deeper understanding of complex social phenomena [41]. Shanghai was selected as the research setting based on the following reasons: Firstly, it was listed in the first batch of megacities that promote participatory regeneration, making the participatory practices in Shanghai diverse, typical, and representative; secondly, it has relatively high-level development of its market and civil society sectors compared to other Chinese cities, making the exploration of collaboration among different stakeholders possible and meaningful; thirdly, it has issued municipal policies to enhance social governance and neighborhood development, making the aim of achieving collaborative governance targeted and pointed; finally, due to facing severe challenges against rapid urbanization, heritage neighborhoods in Shanghai experience tensions between conservation and the need for living situation improvements, making the dynamic of participatory strategies useful and referential for other cities.

Cases were selected based on the following four criteria: (1) neighborhood regeneration projects with participatory strategies; (2) projects involving different multi-level

stakeholders; (3) neighborhood regeneration projects that were finished within three years, to ensure the availability of witness residents and key stakeholders; and (4) projects covering different typical modes of participatory neighborhood regeneration in heritage neighborhoods. Hence, three cases were selected, including the planner-dominated GX neighborhood regeneration (Case A), the planning authority-dominated DX neighborhood regeneration (Case B), and the planner well-organized ST neighborhood regeneration (Case C).

### 3.2. Measures

To fulfill the first objective of the investigation of collaborative conditions associated with participatory regeneration projects, the following four aspects were measured based on the previous literature: participative decision-making, inclusion, collective problem-solving, and power-sharing. Specifically, participative decision-making, relating to the decision makers and decision-making processes achieved through participants' behaviors and activities [29], was measured through authoritative power relations, administrative structures, the empowerment of participants, the bureaucratic control of the decision makers, the content of the regeneration project, and the selection of the final planning strategies [30]. Inclusion, relating to the wide range and multi-level types of participants [24], was measured via the strategy of involving participants, the selective strategy of representatives, participants' activities during the participating process, and to what extent their voices were heard [33]. Collective problem-solving, relating to the collaboration among participants in achieving their shared vision and commitment to problem-solving [34], was measured via the identification of efforts through which the collective problems were tackled and the collaborative mechanisms among these participants. Power-sharing, relating to the equilibrium of power arrangements and corresponding responsibilities in a setting of participation [39], was measured via the formal hierarchical administrative relations, the informal dispersing arrangements, and the temporary sharing of financial relations among multiple levels of stakeholders [38].

To fulfill the second objective of the investigation of determining the facilitating and constraining mechanisms for shaping collaborative governance conditions, the key features associated with the regeneration projects were summarized and analyzed, covering physical improvements, social participation activities, and institutional arrangements. Physical improvements in the built environment, consisting of collectively owned open and accessible places [42], were measured via the size, function, and architectural and landscape design of the regeneration projects. Social participation, relating to participation in social activities rather than political voting, was measured via participation in social activities, membership in social associations or groups, and volunteering [26]. Institutional arrangements, demonstrating the setting for mobilizing and incentivizing participation through procedures and regulations [27], were measured via formal administrative regulations, the participation procedure, incentive mechanisms promoted by the government and public sector, and informal strategies raised by market and civil society entities.

### 3.3. Data Collection

This study employed multiple data sources, covering participatory site observations, in-depth interviews, and documentary materials (e.g., government and public department reports, internal documents, and online media) to reduce the bias generated by using a single source. Specifically, to capture the process and outcomes of participatory regeneration, site observation was conducted in the regenerating communal space during the afternoons on weekdays and weekends, aiming to capture the key features and the usage of the places, and participatory site observation was employed in regeneration project-related workshops, consultation meetings, and other organized activities. In-depth interviews were conducted from March 2016 to December 2021 by the co-authors, with the aim of gaining a better understanding of how participatory regeneration affected participative decision-making, inclusion, collective problem-solving, and power-sharing, as well as

to clarify similar and different mechanisms affecting the shaping of collaborative governance in three cases. The interviewees included stakeholders (e.g., professionals with rich practical experience or sufficient knowledge and key actors involved in the projects) and residents (e.g., 10 respondents in each case were selected through snowballing sampling starting from the planners, and other resident interviewees were randomly selected in the communal space), acquiring 17 key stakeholders and 45 residents in total (Table 1). All interviews were recorded and transcribed, ranging from 20 to 60 min in length, and the total time of recorded interviews was around 35 h. The questionnaire is included in Appendix A. Interview data were collected without involving any community organization personnel to ensure that respondents' answers were not affected, and all participants were informed of their right to withdraw at any time. Also, the co-authors obtained various documentary evidence, including policy regulations and documents, public department internal reports, meeting notes, committee reports, media outlet coverage, government press releases, and news from social media. These materials were supplementary and helped us to better understand and interpret the first-hand data.

**Table 1.** Interviewees.

| Types | Number | Interviewees |
|---|---|---|
| Local officials | No. 1 | One staff member from the Street Office |
| | No. 2 | One staff member from the Department of Housing Management |
| | Nos. 3–4 | Two staff members from the Planning and Natural Resource Bureau |
| | No. 5 | One staff member from Public Space Promotion Center, affiliated with the Planning and Natural Resource Bureau |
| Experts | Nos. 6–9 | Four professors from local universities who were experts in participatory neighborhood regeneration |
| Planners | Nos. 10–12 | The three community planners in three case studies |
| Leaders from community organizations | Nos. 13–15 | The three leaders of Residents' Committees in three case studies |
| | Nos. 16–18 | The three leaders of Property Management Companies in three case studies |
| | Nos. 19–21 | The three members of community interest groups |
| Residents | Nos. 22–38 | Seventeen residents from Case A (the GZX neighborhood) |
| | Nos. 39–54 | Sixteen residents from Case B (the DX neighborhood) |
| | Nos. 55–66 | Twelve residents from Case C (the ST neighborhood) |

*3.4. Data Analysis*

Since this study aimed to explore facilitating/constraining mechanisms, which could hardly be generated from existing theoretical frameworks in the literature, the application of inductive reasoning was vital to obtain ideas or concepts when reviewing and coding qualitative data. Starting from designed questions, a grounded theory method was employed to systematically analyze the interview data, due to its beneficial capabilities of constructing theories from the data themselves and discovering theories from the data of a particular phenomenon [43–45]. In this study, we adopted multi-layer data coding guidelines (involving open coding, axial coding, and selective coding), scrutinized data by continuous comparison, wrote memos (covering codes, emerging categories, categories linkages, features and issues, and ideas), and integrated the generated theoretical links into existing knowledge [46]. The analysis proceeded in 3 steps. Step I was coding initial data and grouping initial data into first-order categories which could demonstrate interactions among all stakeholders, containing grassroots government initiated, market involvement, residents, empowerment, negotiation, commitment, physical improvements, and conflicts among stakeholders. Step II was listing the axial coding, with the aim of looking for core themes to explain the four conditions for achieving collaborative governance in each case,

and classifying first-order categories into second-order themes, aiming to capture different stakeholders' participation and collaboration. For example, the second-order themes of active government, active public department, government support, and government supervision under the first-order category of grassroots government initiated. And the second-order themes were selected residents, resident representatives, and nominated residents under the first-order category of residents. Step III was theorizing and generalizing these themes, aiming to explain the influencing role of participatory regeneration in shaping collaborative governance conditions in future.

Based on this coding and comparing process, key features were summarized and identified to capture participatory decision-making, inclusion, collective problem-solving, and power-sharing dimensions. In the participatory decision-making dimension, the key features were the planner-dominant decisions in integrating selective comments into the final plan (Case A), the planning-authority dominant temporary decision-making committee (Case B), and the planner-designed decision-making meetings (Case C). In the inclusion dimension, the key features identified were the Residents' Committee's selective inclusion (Case A), the planner authority-regulated inclusion (Case B), and the planner's selective inclusion (Case C). In the collective problem-solving dimension, the key features were the leader of the Residents' Committee with strong resource mobilization capacities (Case A), the final plan being selected by a temporary jury committee (Case B), and the planner-integrated stakeholders' comments based on a series of consultation meetings (Case C). In the power-sharing dimension, the key features were sharing landscape maintenance responsibilities with the volunteer group (Cases A and B) and sharing the communal space maintenance responsibilities with the nearby residents and sharing the funding provision with a private company (Case C).

## 4. Findings

### 4.1. Case A

The Case A neighborhood was built in the 1920s, with 780 households. Most residents were tenants (e.g., retired workers and immigrants) and had to share kitchens and toilets with their neighbors. The planner was an urban design professor from a local university and was invited to regenerate this neighborhood by the leader of the Street Office (the grassroots government, which is a sub-level of the district government) because of his excellent achievements and reputation in Shanghai. Owing to the dilapidated built environment and poorly managed communal space conditions, the participatory regeneration project was conducted and completed in 2017, including updates to the leisure facilities (e.g., seats, fitness equipment, laundry shelves, and flower shelves), the reorganization of existing functions into integrated spaces, and the adding of communal functions (e.g., a shared living room, a reading corner, and a study for pupils).

In terms of participatory decision-making, the issues in this neighborhood were identified by the planner, and so was the content of this regeneration, which was based on a detailed site investigation and a residents' survey. "The official from the Street Office asked us to fully support the planner to promote and implement the project. The planner proposed his plan to residents' representatives in we organized consulting meetings. If there were opponents, I needed to explain and persuade these opponents not to against the project" (Interviewee No. 13; 18 March 2016). Based on this interview, we found that the decisions of regeneration (the content of the regeneration and the decision on the final plan) were made by the planner, even though the final plan was developed based on the residents' survey and some resident representatives' comments. Although some residents' comments were incorporated into the final plan, comments that could hardly be integrated relied on the persuasions of and interactions between opponents and the leader of the Residents' Committee. In this regard, resident representatives, neighborhood organizations (e.g., the Residents' Committee and property management company), and officials from Street Office and Housing Management Department participated in the decision-making through attending Residents' Committee-organized meetings, but only

selective residents' comments were integrated into the final plan, making this planner-dominated selective decision-making.

In terms of inclusion, resident representatives were included through their attendance at consultation meetings. It is noteworthy that most resident representatives were building or floor leaders, who normally were retired workers from state enterprises (*danwei*) with long lengths of residence and who normally had a close relationship with the Residents' Committee. Based on interviews, similar situations were also observed regarding inclusion in Case A. "We adopted participation in regeneration and invited residents to give comments. Owing to limited time and energy, we asked the Residents' Committee to organize consulting meetings and invited residents. Although all residents who attended the consulting meetings agreed with our plan, there were still opponents against the plan when implementing the project" (Interviewee No. 10; 15 March 2017). In this case, resident representatives were neither elected by residents nor selected by the planner but were nominated by the Residents' Committee due to their close relations, meaning that these resident representatives could hardly reflect most residents' opinions within the neighborhood. In this aspect, the inclusion of participants was biased and quite limited, and no further inclusiveness mechanisms were adopted to make the participatory regeneration more inclusive.

In the dimension of problem-solving, since the communal space was quite limited and residents did not move out during the regeneration projects, most conflicts associated with the regeneration were due to either changes being made to their previous communal space usage habits (e.g., changing the orientation of the toilet and changing a previous activity room to have new functions) or complaints about the regeneration disturbing their daily lives (e.g., occupying their laundry-hanging spaces and indiscriminately moving parking bikes and electric motorcycles to the designated site, etc.). Also, since most residents were retired workers with relatively low incomes and were, therefore, incapable of improving their living standards alone, they relied greatly on the Residents' Committee (serving as the administrative agency of the government) and the Housing Management Department to deal with their housing issues, there were some opponents who were against this project, because they wanted the neighborhood to be demolished so that they could be compensated and relocated to a spacious apartment, and there were some opponents who wanted to take this opportunity to update their private space or facilities using public funding (Interviewee Nos. 22–23–27; 9–10 June 2018). "For the comments the planner did not integrate into the final plan, I had to mobilize my reputation and personal networking to persuade the opponents" (Interviewee No. 13; 18 March 2018). In this case, their reliance on the resource mobilization capacity of the leader of the Residents' Committee meant that the problem-solving mechanism, in this case, could hardly be transferred to and duplicated in other neighborhoods.

In the power-sharing dimension, owing to the Street Office leader's nomination, the planner invited communal-space-related stakeholders to the consultation meetings, including the leader of the Residents' Committee, the leader of the property management company, staff from Street Office, staff from the planning authority and the Housing Management Department, and resident representatives. After the regeneration, the Residents' Committee was responsible for the management of the regenerated dining room, and the existing volunteer group was responsible for the management of the greenery located on the added shelves in the central square. In this case, the power to determine the content and the final plan of the regeneration was shared by the Street Office and the planner, and the benefits of renting out the space by the hour belonged to the Residents' Committee, the money from which was used for the operation and maintenance of the communal space.

In brief, Case A involved selective participatory decision-making, the inclusion of key actors and Residents' Committee selective resident representatives, and a kind of power-sharing, but the collective problem-solving relied on the leader of the Residents' Committee, which hardly affected the governing mode of this neighborhood and hardly contributed to the transformation towards collaborative governance (Figure 2).

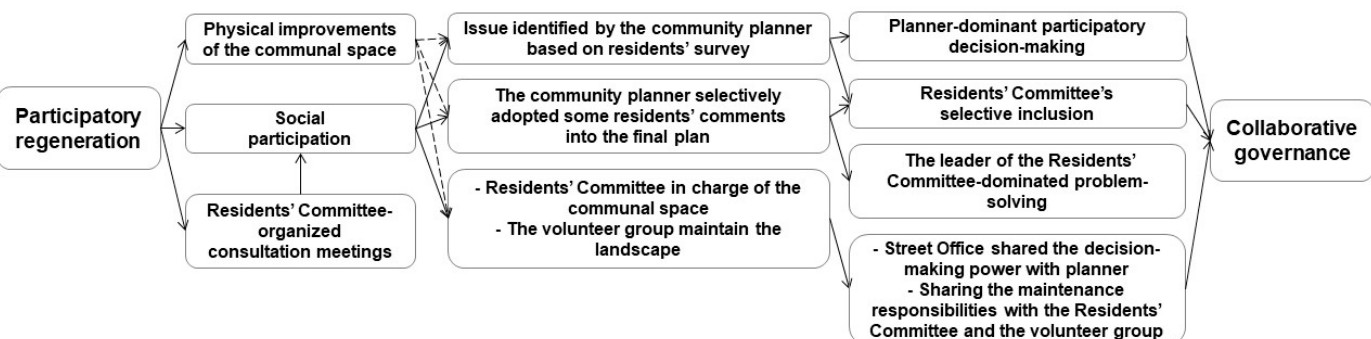

**Figure 2.** Elements of participatory regeneration in achieving collaborative governance (Case A).

*4.2. Case B*

Case B, a terraced house neighborhood with a garden and a large area of poorly managed landscape, was built in the 1930s, with 420 households. Most residents were tenants (including retired workers and their family members, immigrants, etc.). In this case, due to a lack of laundry-hanging facilities and spaces for social activities, the regeneration project added daily necessary facilities (e.g., seats, fitness equipment, and laundry shelves), integrating daily service functions with landscape design (e.g., two small squares surrounded by landscape) in 2017.

In terms of decision-making, the content of the regeneration was proposed by the planner based on a site investigation and a residents' survey, and the decision on the selection of the final plan (out of three to five alternative plans) was determined by a temporary multi-party jury committee regulated by the planning authority, which included staff from the planning authority, officials from the Street Office, the leader of the Residents' Committee, professional experts in practice, staff from the property management company, and two resident representatives selected by the Residents' Committee. "I joined the jury committee in this neighborhood regeneration, there were different standpoints from different stakeholders. Street Office cared more about finishing the project with provided funding before the close of the financial year. Experts cared more about the rationality of the design and reasonable of the functions, planning authority supervised the project conducted without involving property right conflicts and cooperated with other public department if necessary" (Interviewee No. 3; 24 May 2018). In this case, a regulated temporary multi-party jury committee was created for decision-making, serving as a useful and realizable participatory decision-making mechanism for future collaboration.

In terms of inclusion, apart from professional experts and planning authority officials, other stakeholders related to the communal space were included in the temporary multi-party jury committee regulated by the planning authority. These stakeholders attended the consultation meeting, giving comments and voting for the best plan from their standpoints. "We tried to absorb comments from different aspects, including resident representatives, property management companies, residents' committees, officials from the Street Office and professional scholars. Admittedly, the voices from planning authorities and scholars were relatively strong, and normally only two residents attended and nominated by the Residents' Committee" (Interviewee No. 4; 24 May 2018). In this case, although residents were included in the temporary multi-party jury committee who could give comments and vote, their numbers were few and their selection was biased, making the residents' voices weak.

In the problem-solving dimension, the temporary multi-party jury committee was only responsible for the selection of the planner and the final plan; there were no other problem-solving mechanisms to deal with conflicts associated with the regeneration project. In addition, in the dimension of power-sharing, since the regenerated communal space did not involve operation or management, no other group or organization shared the power of management. As previous managers of the communal space, the Residents' Committee

was still in charge of the regenerated communal space, meaning that the power-sharing in this case was quite limited.

In brief, this case involved participative decision-making and the inclusion of stakeholders through the temporary multi-party jury committee, which was a planning authority regulated procedure for promoting participation. This may have facilitated participation, cooperation, and negotiation among stakeholders, but the residents' voices were weak (Figure 3).

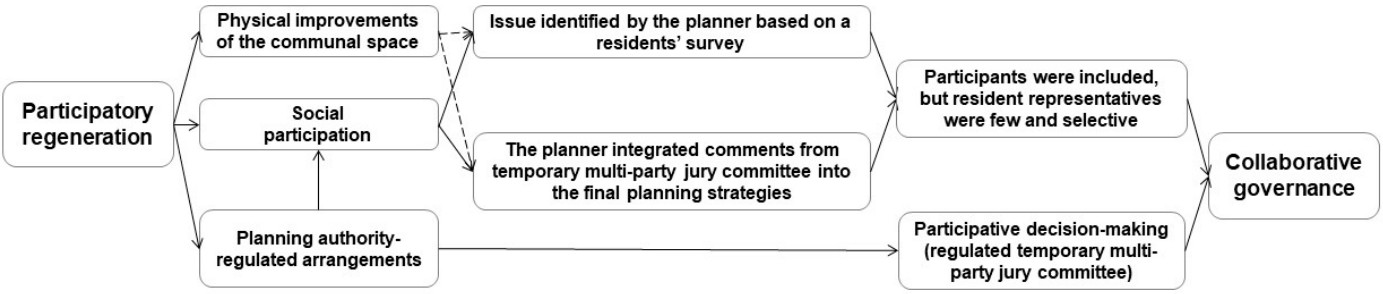

**Figure 3.** Elements of participatory regeneration in achieving collaborative governance (Case B).

### 4.3. Case C

Case C, a terraced house neighborhood with a garden and 168 households, was built in the 1920s. Most residents were tenants. The neighborhood was long and narrow with quite limited communal space and only one entrance, and there was a small underused fitness yard located at the end of this neighborhood. The first round of the participatory regeneration project was conducted in 2018 and finished in 2019, which included transforming the small fitness yard into a multi-functional open space with a folding rain-shed, and adding seats equipped with plant shelves, rainwater collection mechanisms, and laundry racks. The second round of the participatory regeneration project was conducted in 2021, which included updating the structures and the folding roof.

In terms of the decision-making, the contents of the regeneration project were gradually developed through a series of planner-organized consultation meetings, and the planner invited both residents and other stakeholders to attend. "At first, the Residents' Committee invited me to design the guidepost and doorplate system, but I find the communal space lack vitality and social activities which stimulate my organization of consulting meeting to help residents in this neighborhood to identify issues" (Interviewee No. 12; 12 June 2018). The first meeting was held to collect residents' opinions on collective issues, especially communal-space-related issues. Two key issues were identified: the lack of laundry racks and the lack of open space. Then, the second meeting was held to discuss how to improve the communal space to meet these needs and to raise suggestions for the regeneration project. Then, the third meeting was held, in which the planner proposed planning strategies and designs to the participants and residents and other stakeholders gave opinions and suggestions. After several consultation meetings, the final plan was confirmed. In this case, the participative decision-making was achieved through attending a series of planner-organized consultation meetings, which could be viewed as the first facilitating mechanism for achieving future collaboration.

In terms of inclusion, participants attending the consultation meeting included the leader of the Residents' Committee, officials from the Community Development Office of the Street Office, staff from the property management company, resident representatives, and representatives from a private company. Resident representatives were informed and invited by the leader of the Residents' Committee, and the private company was invited by the planner, owing to his wish to attract private funding to support this project. Although the inclusion of participants was selective, many resident participants were included and most key stakeholders were included, making the participation of this project quite inclusive.

In the problem-solving dimension, a series of consultation meetings were set up to guide residents to identify issues with the communal space, and residents' and stakeholders' comments were integrated into the final plan. However, this project could not obtain funding support, either from the Street Office or from the Residents' Committee. This was because it was not initiated by the grassroots government and, therefore, could not obtain financial support through public funding. "After all participants satisfied with the final plan, I had to try my best to implement it. After discussion with officials from Street Office and the leader of the Residents' Committee, I talked with several private companies and invited them to support this project. Finally, one top global 500 companies agreed" (Interviewee No. 12; 12 June 2018). In this case, when facing problems, such as identifying communal-space-related collective issues and looking for funding support, the planner had to mobilize his resources, together with taking suggestions from officials from the Street Office and the Residents Committee, which were more familiar with resources near this neighborhood.

In the dimension of power-sharing, the planner suggested setting up a volunteer group to manage the regenerated communal space, especially the folding roof. However, owing to the retirement of the previous leader of the Residents' Committee, the volunteer group was not set up. One day, there was heavy rain, but no one helped to fold the roof, meaning too much water had been kept on the roof overnight, which deformed one of the key structural frames and some supporting components. Although the key structural frame was changed and other frames were repaired to their previous state, the desirable folding–unfolding operation did not work afterwards, leaving the well-designed roof folded and underused for years. In 2021, this communal space experienced a second round of regeneration for the folding roof. Based on collaboration with the Residents' Committee, who communicated with residents, and the private company, who provided the funding, the planner regenerated the roof into a more stable and easy-to-operate new one and assisted the Residents' Committee in establishing a volunteers' group to maintain the communal space. In this case, the planner shared the power of regenerating the communal space based on the acquirement of private funding, and the volunteer group shared the power of managing the folding roof with the Residents' Committee. In general, these elements of participatory regeneration in achieving collaborative governance were shown in Figure 4.

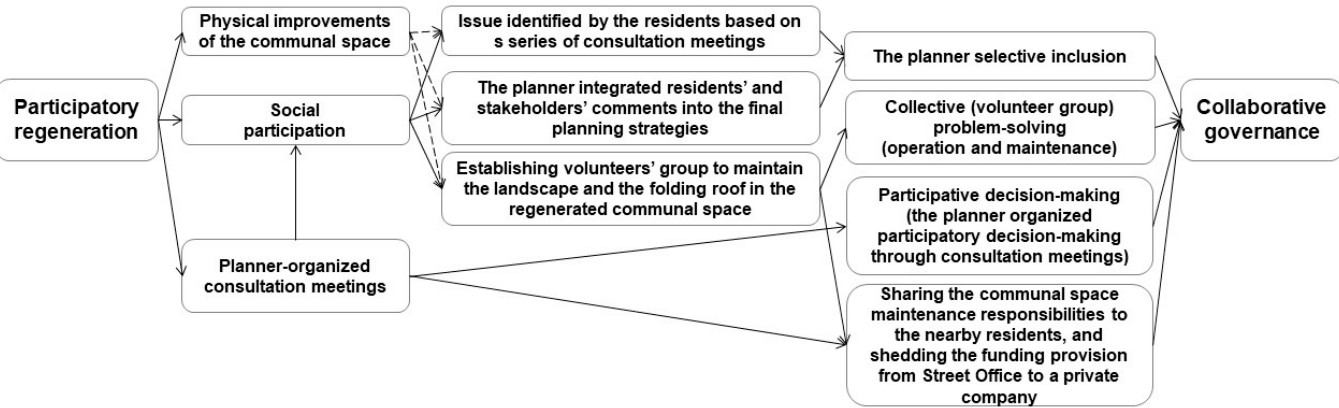

**Figure 4.** Elements of participatory regeneration in achieving collaborative governance (Case C).

In brief, key elements associated with participatory regeneration relating to the conditions for achieving collaborative governance were observed in all three case studies with nuanced differences (Table 2). This study found that (1) the planning authority regulated participation procedures and the planner organized consultation meetings tend to create highly participatory decision-making; (2) different inclusion strategies were observed and generated different inclusiveness of residents and actors; (3) only Case C involved collective problem-solving after the regeneration project, due to regularly organized consultation

meetings; and (4) not only state administrative nomination but also financial resource mobilization tend to create power-sharing between stakeholders.

**Table 2.** Key elements associated with participatory regeneration in achieving collaborative governance in three case studies.

| Cases / Dimensions | Case A | Case B | Case C |
|---|---|---|---|
| **Participative decision-making** | **Low** (The planner proposed the plan and integrated selected residents' comments into the final plan) | **High** (Based on the planning authority regulated participation procedures) | **High** (The planner organized consultation meetings) |
| **Inclusion** | - Residents' representatives nominated by the Residents' Committee; <br> - The property management company; <br> - The Residents' Committee; <br> - Officials from the planning authority and the Housing Management Department. | - Officials from the planning authority, the Street Office, and the Housing Management Department; <br> - The leader of the Residents' Committee <br> - Staff from the property management company; <br> - Resident representatives. | - Resident representatives; <br> - Leader of the Residents' Committee; <br> - Staff from the property management company; <br> - Officials from the Street Office; <br> - A private company. |
| **Collective problem-solving** | **No** (Relying on the reputation and personal social networking capabilities of the leader of the Residents' Committee) | **No** (No collective problem-solving arrangements after the decision of the final plan) | **Yes** (Based on regularly organized consultation meetings) |
| **Power-sharing** | **Yes** (The Street Office shared the power with the nominated planner; the volunteer group shared the power of management) | **No** | **Yes** (The Street Office shared power with the planner owing to the acquirement of the funding; the volunteers shared the power of management) |

## 5. Discussion: Affecting the Conditions of Collaborative Governance

### 5.1. Facilitating Mechanisms

The first facilitating mechanism was the employment of consultation meetings, including a public hearing, a coordinating meeting, and a convocation, with more meetings that could be organized if necessary. This consultation meeting system was promoted by the Shanghai Municipal Civil Affairs Bureau in 2006 and was codified in the "Working Regulations of Residents' Committees in Shanghai" in 2017, which attempted to codify the organization of a public hearing before the event, a coordinating meeting during the implement, and a convocation after the event. "Although the consultation meetings were regulated, not all Residents' Committees make the most of them" (Interviewee No. 3; 23 May 2018). The planner organized a series of consultation meetings, could be viewed as another effective approach facilitating mechanism for achieving future collaborative governance, apart from community planner organized collaborative workshops associated with participatory planning [18].

The second facilitating mechanism was the regulated participation procedures announced by the planning authority when promoting "Micro Communal Space Regeneration Projects" in 2016. The procedures were as follows: (1) calling for pilot neighborhoods to propose the issues and expectations; (2) organizing officials from the Street Office and the planning authority and professionals to collectively select appropriate neighborhoods for regeneration; (3) calling for planners to raise proposals; (4) selecting proposals based on the decision made by a temporary multi-party jury committee; (5) deepening the proposal and encouraging participation in the regeneration. Although the vital role of community

responsible planner institutional system in power construction has been confirmed using cases from Beijing, China [47], this study highlights public-sector-regulated guidelines and procedures in participation that are still important for achieving collaborative governance.

### 5.2. Constraining Factors

The first constraining factor is rooted in administrative arrangements. The planning authorities in question did not have the capacity to mobilize to the same level of other public departments (e.g., the Housing Management Department or Landscaping and City Appearance Administrative Bureau), and, compared with other lateral public departments, most neighborhood-based communal-space-related issues could not merely be solved by planning strategies because of the issues with property rights and relocating neighborhood-based facilities and resources. While, currently, most participatory regeneration in Shanghai is promoted by the planning authority, having collaborations between different public sectors is inefficient and difficult, especially when facing property rights and the adjustment of land use challenges.

Most grassroots state-led participatory regeneration highly relied on public funding provided by the grassroots government. In Case A and B, the Street Office provided the funding and supervision strategies (i.e., they nominated the planner in Case A and served as core members of the temporary jury committee in Case B), leaving limited space for residents and private actors. Projects using public funding needed to follow strict financial approval conditions and schedules, and normally the funding was provided annually as a one-off payment, meaning the project managers had to square the account before September that year, which was normally mismatched with the schedule of the participatory regeneration project. Also, this financial arrangement did not include the operation costs, management fees, and maintenance expenses after regeneration.

### 5.3. Uncertainties

The subjective initiatives of the leaders of the Residents' Committees, the responsible officials from the Community Development Office of the Street Office, and planners with strong resource mobilization capabilities played vital roles in facilitating the participatory neighborhood regeneration projects. For example, in Case C, the success of the participatory regeneration project also relied on the collaboration between the director of the Community Development Office of the Street Office, who supported and promoted the project, the leader of the Residents' Committee, who was good at mobilizing and facilitating residents' participation initiatives, and the community planner, who could mobilize social resources and with professional skills. Their cooperation was crucial for the effectiveness of participatory regeneration, but these conditions were hard to achieve, owing to the difficulties in transferring to or duplicated in other neighborhoods.

### 6. Conclusions

Many studies have explored the role of existing collaborative governance [3,19,21] or collaborative workshops [18] in promoting participatory regeneration. However, this study highlights the dynamic roles of participatory planning and management strategies in shaping future collaborative governance, and we identified facilitating mechanisms (e.g., consultation meetings, regulated participatory procedures, and flexible private funding provision), constraining mechanisms (e.g., administrative constraints in mobilizing other lateral public departments and strict arrangements for public funding provision), and uncertainties (e.g., responsible officials, capable leaders, and planners with strong resource mobilization capabilities) in China's urban heritage neighborhoods with strong state interventions. This study deepens the identification of nuanced influencing mechanisms associated with participatory regeneration in achieving collaborative governance, and it emphasizes the efforts that should be made in strengthening facilitating mechanisms and weakening constraining mechanisms for effective neighborhood service delivery and collective goods management, the efforts that should be made in providing sustainability tools

through facilitating participation in decision-making and solution-solving, and efforts that should be made in enhancing inclusion, through involving diverse residents, and in power-sharing, through collaborations between stakeholders. In this regard, these findings guide participatory regeneration towards not only providing appropriate facilitating mechanisms for achieving collaborative governance when dealing with collective issues but also towards building social sustainability when cultivating inclusion and collective service delivery in future. Furthermore, our analysis underlines the importance of distinguishing different participatory modes and provides suggestions for promoting corresponding strategies to overcome constraining and uncertain factors when developing social sustainable developments and achieving collaborative governance in different scenarios. In planner-dominant regeneration, the capable leaders of the Residents' Committees were highly important for problem-solving, which was effective but did not contribute to the conditions for achieving collaborative governance. In planning authority-dominated regeneration, the facilitating mechanisms were the regulated procedures for integrating participation into the different steps of the regeneration projects, in particular, the outlines for the organization of multi-party jury committees for collective decision-making. In planner-organized regeneration, the availability of funding and the integration of different steps with consultation meetings were vital mechanisms for achieving collaborative governance. In addition, the cultivation of responsible neighborhood organizations for maintenance and the training of planners to acquire funding support were vital factors for achieving collaborative governance, so relevant guidelines or regulations to this end are suggested. Although this study examines heritage neighborhoods in Shanghai, these results and implications may be generalizable to other neighborhoods with significant cultural and social values in cities with strong state interventions. Limitations need to be acknowledged. The first limitation relates to the research scope. Achieving a more profound knowledge on the influencing role of participatory regeneration would require investigations into other neighborhood types in other contexts. The second limitation relates to the sample size of the interview data, and if more interviews were conducted, the findings could be more representative. The third limitation relates to the interviewees' bias, because they were selected based on the combination of snowball sampling and selective sampling (searching for residents in the communal space rather than door-to-door household survey). Further research could involve expanding the research scope into other neighborhood types which face collective issues and employing participatory regeneration approaches, expanding the breadth of interviews to include more residents and deliver more convincing evidence.

**Author Contributions:** Conceptualization, Y.X.; methodology, Y.X.; investigation, H.W.; resources, Y.X.; writing—original draft preparation, Y.X.; writing—review and editing, H.W.; visualization, H.W.; supervision, Y.X.; project administration, H.W.; funding acquisition, H.W. All authors have read and agreed to the published version of the manuscript.

**Funding:** This research was funded by the Shanghai Art Science Planning Project (No. YB2023-G-092) and the Fundamental Research Funds for the Central Universities (No. 2232022E-10).

**Institutional Review Board Statement:** Not applicable.

**Informed Consent Statement:** Informed consent was obtained from all subjects involved in the study.

**Data Availability Statement:** The data presented in this study are available from the corresponding author upon reasonable request.

**Acknowledgments:** The authors thank those who participated in this research and contributed their valuable time to support this study.

**Conflicts of Interest:** The authors declare no conflicts of interest.

**Appendix A. Questionnaire**

(1) How did you participate in the decision-making process, who played important roles, were there factors or mechanisms strengthening or weakening the participatory decision-making?

(2) What kinds of residents and stakeholders were included in this project, why they are included, what kinds of resources they could mobilize, and how did they collaborate?

(3) How did the project solve collective issues, who played vital roles and in what ways?

(4) Who played dominant roles in the planning stage, management stage, dealing with conflicts associated with this project, respectively, what were administrative or financial or collaborative relations among actors and in what ways?

(5) Generally, what factors or mechanisms did you think might facilitate or constrain the planning and implementation of this projects, and why?

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
