# Peer review of "Achieving Neighborhood-Level Collaborative Governance through Participatory Regeneration: Cases of Three Residential Heritage Neighborhoods in Shanghai"

_sustainability, doi:10.3390/su16052082_

Round 1

Reviewer 1 Report

Comments and Suggestions for Authors

Dear Authors,

Thank you for the interesting article which contributes to the scope of Sustainability. I have only some comments below, and I recommend proofreading:

- line 24: delete "vivid Chinese" as it is unnecessary

- line 25: replace "where are" with "which are"

- line 48: explain the "hukou" system

- line 59: change "Under such circumstance" to "Under such circumstances"

- line 213: “s for” - “as for”?

- lines 271-276: Clarify who conducted the interviews and provide information about the interviewers' background. Also, mention the total time of recorded interviews. Your questionnaire(s) could be included in Appendix.

- line 284: since you show the interviewees' numbers (or IDs) in Table 1, consider referencing their citations in the main text using those numbers. For example, the citation on lines 324-328 could be attributed to interviewee no. 13, 14, or 15.

- Figures 1, 3, 4 & 5: check if the resolution is good enough as the text appears blurry.

- lines 413, 416, 424 & 427: correct "muti" to "multi"

- line 469: clarify what is meant by "nearby private." Is it a private company? If so, line 470 should be revised to refer to "the private company."

- line 505: change "Figure 1" to "Figure 5"

- lines 524-525: “<Working … Shanghai>” - How about changing it to: 'Working ... Shanghai'?

Comments on the Quality of English Language

There are some typos that need to be corrected.

Author Response

Dear Reviewer,

Thank you for reviewing this article. I have made revisions based on your comments and proofreading was done.

Deleting “vivid Chinese” in line 24

Replacing “where are” with “which are” in line 25

Explaining “hukou” system in line 48

Changing “under such circumstance” to “under such circumstances” in line 61

Changing “s for” to “as important for” in line 234

Clarifying who conducted the interviews in line 313

Questionnaire is included in appendix in line 710-720

Changing referencing interviewees citations using those numbers in line 395, 416,437-440, 482, 494, 528-529, 557

Checking the resolution of figures

Correcting “muti” to “multi”

Revising “nearby private” to “the private company” in line 542-543

Changing “<Working … Shanghai>” into ‘Working … Shanghai’ in line 596

Reviewer 2 Report

Comments and Suggestions for Authors

Dear Authors:

Research into "heritage residential neighbourhoods". However, it seems to me that dissociating this issue from sustainability is a major shortcoming. Furthermore, it is important that the conceptual framework be based on more up-to-date references and frameworks, such as SCOPUS and WOS.

In addition, your manuscript needs to be revisited in relation to the following aspects:

-Definition of hypotheses and research objectives.

- Adequacy of the methodology (Participants, Instruments, etc.).

I give "3.3 Data collection" as an example: it lacks scientific consistency (supporting theoretical framework). I ask: how was the participant observation based on what theoretical framework? How was the interview structured? Why the number of participants? Was the research approved by an Ethics Committee?

- Alignment of the methodology to the research objectives.

- Results that respond to the planned research objectives. 

For example: What are the categories for analysing the results?

- Clarity in the way expected results are described and interpreted.

- Relation to previous studies. Limitations. Future lines of research. Clearly stated conclusions. 

Note: Figure 2. does not seem to be of interest. It doesn't add any relevant information.

Rew

Author Response

Dear Reviewer,

I have made revisions based on your comments and proofreading was done.

Adding introduction and discussion on the relationship between this study and sustainability in lines 31-44 and 659-675

Adding more up-to-date references in conceptual frameworks in lines 161, 163,171,200-201,206

Defining the hypotheses and research objectives in lines 126-131

Adding methodology in lines 242-246, 309-312, 317-324, 334-341, 345-358

Aligning the methodology to the research objectives in lines 269-270, 290-292

Results that respond to the planned research objectives in lines 345-358

Clarifying the results in lines 580-588

Relations to previous studies in lines 650-652

Limitations in lines 689-696

Future lines of research in lines 696-699

Clearly stated conclusions were revised in lines 659-675

Reviewer 3 Report

Comments and Suggestions for Authors

Comments on the Quality of English Language

The writing style is fine, careful proof reading is required and some sentences appear to be inclomplete.  See attched. 

Author Response

Dear Reviewer,

I have made revisions based on your comments and proofreading was done.

The introduction was revised by adding foundation of the study, the background information, and more up-to-date reference, justification of the conceptual model, research gap, research questions.

Conceptual framework was developed based on adding more up-to-date references and more analysis based on the literature.

Research design part was modified based on suggestions.

Findings were revised, adding limitations and suggestions for future work.

More up-to-date references were added.

We benefit a lot from your comments, thank you so much for your detailed comments and suggestions.

Round 2

Reviewer 2 Report

Comments and Suggestions for Authors

Dear authors:

The improvements made to the manuscript are relevant. The article is now publishable.

Rw

Reviewer 3 Report

Comments and Suggestions for Authors

The paper is noticably improved.  I wish the authors all the best with their research endeavors.